

# Trade-offs between aquatic and terrestrial locomotion and functional parallelism in *Desmognathus* salamanders

Benjamin M. Fitzpatrick

Department of Ecology and Evolutionary Biology, University of Tennessee—Knoxville, Knoxville, TN, United States of America

## ABSTRACT

A trade-off between aquatic and terrestrial locomotion is self-evident at broad phylogenetic scales. While the effects of more subtle trade-offs in the evolution of closely related species are less clear, they are hypothesized to drive ecological speciation and adaptive radiation. Amphibious animals strike a balance between aquatic and terrestrial activity, and the need to maintain performance in one medium is hypothesized to constrain evolution of high performance in the other (the running-swimming dilemma). Closely related species of *Desmognathus* salamanders partition local habitats along a gradient from mid-stream to stream edge to completely terrestrial. The trade-off hypothesis predicts that these species will differ in relative running vs. swimming performance depending on the relative importance of each mode of locomotion in their niches. Here, I show that primarily aquatic *Desmognathus* ecomorphs are superior swimmers and inferior runners relative to semi-aquatic ecomorphs using paired escape performance trials in aquatic and terrestrial arenas. I measured performance as the velocity of the fast-start response to simulated predator attack. I tested two species of each ecomorph, representing two divergent clades with parallel evolution of aquatic and semi-aquatic species. Notably, the different southern clade ecomorphs have been genetically isolated for millions of years, whereas the northern clade ecomorphs share a recent common ancestor and interbreed regularly. My results showed a negative correlation between aquatic and terrestrial performance, with aquatic ecomorphs being faster swimmers and semi-aquatic ecomorphs being faster runners. While there was a possible trend consistent with faster swimming speeds of northern forms relative to their southern counterparts, the functional differences between ecomorphs were similar in both clades. These results are consistent with the hypothesis that trade-offs between aquatic and terrestrial locomotion have contributed to a consistent pattern of habitat partitioning during parallel speciation.

## INTRODUCTION

According to the ecological theory of adaptive radiation, diversity is produced by divergent selection for disparate phenotypes in different environments (*Schluter, 2000*). Species produced by divergent selection are ecologically differentiated because of performance

Corresponding author
Benjamin M. Fitzpatrick,
benfitz@utk.edu

trade-offs: The phenotypic syndrome that is optimal in one environment or niche is suboptimal elsewhere and *vice versa*. Examples include feeding on different sized seeds in Galapagos finches (*Grant & Grant, 2006*), benthic *vs.* limnetic foraging in three-spined sticklebacks (*Schluter, 1995*), and locomotion on different substrates in lizards (*Garland & Losos, 1994*). However, some studies concluded that functional trade-offs were absent or weak, particularly in within-species comparisons or groups with similar generalist phenotypes (*e.g.*, *Gvozdik & Van Damme, 2006*; *Nauwelaerts, Ramsay & Aerts, 2007*; *Zamora-Camacho, 2023*; *Peron, 2024*).

Aquatic and terrestrial locomotion present different challenges (*Blob et al., 2016*; *Kawano & Blob, 2022*; *Axlid et al., 2023*), and amphibians such as salamanders are often regarded as examples of general-purpose phenotypes that are not particularly good at either because the requirement to maintain adequate function in one medium constrains the evolution of high performance in the other (*Bonett & Blair, 2017*). *Gvozdik & Van Damme (2006)* labeled this constraint the 'running-swimming dilemma,' but found no evidence to support it among *Triturus* salamanders (family Salamandridae). That is, species that were faster on land were not necessarily slower in water. *Desmognathus* salamanders (family Plethodontidae) provide an opportunity to test the hypothesis in a different ecological and evolutionary context.

The dusky salamanders (genus *Desmognathus*) are hypothesized to be an adaptive radiation with variation in habitat use along an aquatic-terrestrial microhabitat gradient (*Tilley & Bernardo, 1993*; *Kozak et al., 2005*; *Bruce, 2011*). Once, each adaptive zone along the gradient was considered the province of one or two species (*Dunn, 1917*; *Dunn, 1926*). Several decades of behavioral and molecular research have revealed that the group is composed of dozens of species, with repeated evolutionary transitions among ecomorphs (*e.g.*, *Tilley, Verrell & Arnold, 1990*; *Kozak et al., 2005*; *Kozak, Mendyk & Wiens, 2009*; *Tilley et al., 2013*; *Camp & Wooten, 2016*; *Pyron et al., 2022*; *Pyron et al., 2025*). An 'ecomorph' describes a syndrome of similar habitat, morphology, and behavior shared by species that are not necessarily closely related (*Williams, 1972*).

The two largest and most aquatic ecomorphs are the semi-aquatic 'black-bellied' form and the aquatic 'shovel-nosed' form (*Dunn, 1917*; *Martof, 1962*; *Petranka, 1998*). They were formerly classified as two species *D. quadramaculatus* and *D. marmoratus*, respectively. However, molecular phylogenetic studies demonstrated that both ecomorphs occur in each of two divergent clades that are paraphyletic with respect to most other species of *Desmognathus* (*Jackson, 2005*; *Kozak et al., 2005*; *Jones & Weisrock, 2018*; *Pyron et al., 2022*). In the southern clade, the aquatic (*D. aureatus*) and semi-aquatic (*D. amphileucus, D. gvnigeuswotli*, and *D. folkertsi*) species are genetically distinct and reproductively isolated (*Camp et al., 2002*; *Jackson, 2005*; *Pyron et al., 2022*). However, in the northern clade, there is either a single polymorphic species (*Jackson, 2005*; *Jones & Weisrock, 2018*), or a complex of closely related hybridizing species (*Pyron & Beamer, 2022*; *Pyron & Beamer, 2023*; *Pyron et al., 2025*) with little correlation between genomic similarity and ecomorph. These studies show that the evolutionary dynamics of the aquatic and semi-aquatic forms have been different between the northern and southern clades, but there is no evidence as to whether or not there are functional differences between species of the same ecomorph.

Both ecomorphs studied here are stream-dwelling and have keeled tails, hardened toe-tips and a prolonged larval phase, features characteristic of the most aquatic *Desmognathus* species (*Petranka, 1998*). However, the aquatic ecomorphs, initially described as a separate genus (*Moore, 1899*), have more broadly finned tails (Fig. 1) and features of the head and internal nares thought to be associated with their aquatic habit (*Martof, 1962*). While they are capable of survival and locomotion on land, they rarely leave large streams, feed primarily on aquatic prey (*Martof & Scott, 1957*), and appear reluctant to feed when removed from water (*Deban & Marks, 2002*). The semi-aquatic ecomorphs are more often found on stream banks, breed in both large and small streams, and forage primarily on terrestrial prey as adults (*Martof & Scott, 1957*; *Davic, 1991*). Thus, the relative importance of aquatic *vs.* terrestrial locomotion differs between the two forms. I hypothesized that the aquatic and semi-aquatic ecomorphs would differ in relative performance according to the running-swimming dilemma (*Gvozdik & Van Damme, 2006*), but that the northern and southern representatives of each ecomorph would be functionally equivalent despite their disparate histories.

## MATERIALS & METHODS

### Experimental subjects

Ten adult semi-aquatic (*D. amphileucus*) and 10 adult aquatic (*D. aureatus*) morphs were collected from a stream near DeSoto Falls Recreational Area, Lumpkin Co., Georgia. These represent distinct species in the southern clade. Twelve adult semi-aquatic (*D. sp.*) and 12 adult aquatic (*D. marmoratus*) morphs were collected from Little Laurel creek along Whitetop Rd., Smyth Co. Virginia. These represent the ecomorphs in the northern clade. The northern semi-aquatic taxa cannot be distinguished visually and either *D. mavrokoilius* or *D. kanawha* could be present at the Virginia locality (*Pyron & Beamer, 2022*). Collections were made in 2010, before the taxonomic revision of *Pyron & Beamer (2022)*. Based on the locality and genetic data of *Pyron et al. (2022)*, the Virginia locality is in or near a contact zone between *D. mavrokoilius* and *D. kanawha*, and it is likely that they interbreed where they co-occur (*Jones & Weisrock, 2018*; *Pyron et al., 2022*). As a pragmatic matter for this analysis, they are lumped collectively as the northern semi-aquatic ecomorph, *Desmognathus sp.* Aquatic ecomorphs were collected by looking under rocks underwater in stream riffles and semi-aquatic ecomorphs were collected by looking under rocks along the edge of the stream or above water in the stream.

All 44 individual salamanders were evaluated for fast-start performance after 8–12 weeks in captivity. The sexes of the individuals were not known. Based on previous studies (*Azizi & Landberg, 2002*; *Fitzpatrick, Benard & Fordyce, 2003*), we intended to test 10–20 individuals per species. In practice, we let the less common ecomorph dictate the sample size for each collection (the aquatic morph in both cases). Individuals with tail damage were not collected. Individuals smaller than 50 mm snout-vent length were assumed to be juveniles and not collected (*Martof, 1962*). The average and standard deviation of the snout-vent lengths for each group, measured at the time of capture, were as follows: 66.5 ± 0.69 mm for southern aquatic, 74.9 ± 1.52 for southern semi-aquatic, 72.2 ± 1.23 for northern aquatic, and 60.6 ± 0.89 for northern semi-aquatic.

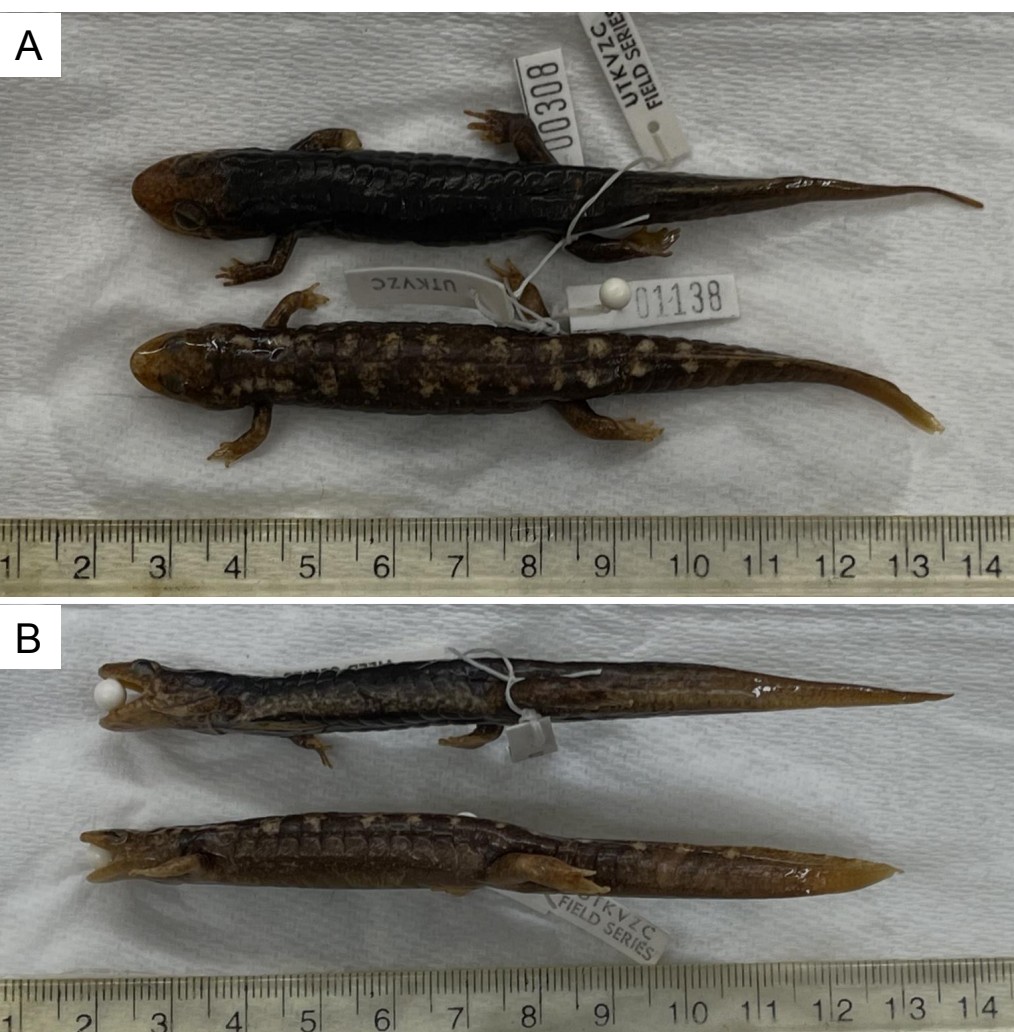

**Figure 1** **Representative specimens of semi-aquatic and aquatic *Desmognathus* ecomorphs.** (A) Dorsal views show the slightly smaller and narrower head of the aquatic ecomorph (lower specimen, closest to the ruler). (B) Lateral views show the more spatulate tail of the aquatic ecomorph (lower specimen, closest to the ruler. Collection data from the University of Tennessee Knoxville Vertebrate Zoology Collection are as follows. Specimen 00308 *Desmognathus amphileucus*: 05 July 1976; West Fork of Mill Branch, NW Slope of Big Fodderstack Mtn, Cherokee Nat. Forest, Monroe Co. TN; 84 02 45 W, 35 25 19 N; 750 m elevation; Map Ref: Whiteok Flats; Collector R. L. Jones. UTKVZC Cat # 00308, Field Number 00116. Specimen 01138 *Desmognathus marmoratus*: 06 May 1977; Tom's Branch, just below mouth of Twin Springs Branch, 5.7 air miles south of town of Roan Mtn. Carter Co.; TN. 82 05 26 W, 36 07 26 N; 3840 ft elevation; Map Ref: Carver's Gap; Collectors W. H. Redmond, R. L. Jones, J. Wojtowicz. UTKVZC Cat # 00831, Field Number 01138. Photo credit: Ben Fitzpatrick.

Salamanders were maintained at 15 °C and 70% r.h. in Sterilite 6-quart (5.7 L) plastic shoeboxes (34.5 × 20.3 × 12.7 cm). Semi-aquatic individuals were housed on a damp paper towel substrate, while aquatic individuals were housed in two L of de-chlorinated and well-aerated water. Bedding was changed weekly and water changed every three to four days. All salamanders were provided with a terra cotta refuge and were fed either crickets

or California blackworms *ad libitum*. Food was withheld for two days prior to performance assessment to minimize effects of recent feeding on locomotion (*Yan et al., 2015*; *Zhao et al., 2020*). Salamanders were monitored daily for signs infection, loss of appetite, or impaired mobility. There were no adverse events.

Experiments were carried out in accordance with the University of Tennessee Institutional Animal Care and Use Committee protocol # 1870-0510, a scientific collection permit from the Virginia Department of Game and Inland Fisheries (# 37891) and the Georgia Department of Natural Resources. All animals were alive and healthy at the end of the experiment and were retained for additional research on mating behavior.

## Escape performance trials

Salamander escape performance was assessed as fast-start velocity in response to a threat stimulus (*Domenici & Blake, 1997*; *Walker et al., 2005*; *Fitzpatrick, 2008*). A few studies have demonstrated that swimming and running velocity are good predictors of survival in amphibians and reptiles (*Watkins, 1996*; *Walker et al., 2005*; *Husak, 2006*). The *Desmognathus* species studied here are more active at night, but can be observed moving about at any time of day (*Martof, 1962*; *Huheey & Stupka, 1967*; *Valentine, 1974*; *Dodd, 2004*). They are not chemically defended and adults are preyed upon by large aquatic predators such as trout (*Salvelinus fontinalis*, *Oncorhyncus mykiss*), and semi-aquatic/terrestrial predators such as snakes (*e.g.*, *Nerodia* and *Thamnophis*), river otters (*Lutra canadensis*), raccoons (*Procyon lotor*), and other medium sized mammals and birds (*Martof, 1962*; *Petranka, 1998*; *Sánchez-Hernández, 2020*; *Dempsey, Roden & Bidwell, 2021*). Thus, the velocity of the initial burst of swimming or running in response to attack is likely an important component of ecological performance. Moreover, sprinting speed can be correlated with other components of performance requiring physical vigor, including foraging and territory defense (*e.g.*, *Peterson & Husak, 2006*).

Escape response was recorded three times for each individual salamander in both a terrestrial and an aquatic arena. The terrestrial arena consisted of a glass container with a gravel substrate and water level with the substrate. The aquatic arena consisted of a glass container with enough water to fully submerge a salamander. Salamanders were given at least 30 min to acclimate to room temperature because the camera setup was in a different room from where the salamanders were housed. The temperature in the camera room was 21–22 C and preliminary tests indicated that 200g of water equilibrated within 30 min. Previous work demonstrated that salamanders typically equilibrate to a 6–7 C change in temperature within 15 min (*e.g.*, *Hutchison, 1961*; *Lunghi et al., 2016*).

After acclimating, a salamander was placed in an arena and allowed to acclimate until it ceased moving. A gentle pinch at the base of the tail with forceps was used to stimulate an escape response. I had no way of measuring the strength of the stimulus. Animals were always approached from above and behind, and presumably they could see the forceps approaching. If they moved before being touched, the forceps were withdrawn and the stimulus was attempted again when the animal stopped moving. On the rare occasions when this occurred three times in a row, the trial was stopped and the animal was moved to the end of the list of animals to be tested that day. Variation in the stimulus

and/or motivation may affect escape responses (*Barry & Harrison, 1957*), but I assume this variation was unbiased with regard to the comparisons of interest for this study.

Escape responses were recorded at a rate of 500 frames per second using a high speed digital camera (NAC Memrecam Ci digital system). Trials were repeated only after at least a 30 min break to reduce stress on the animals and to prevent habituation to the stimulus. *Azizi & Landberg (2002)* found no evidence of fatigue across five escapes during 15 min trials with salamanders, so I assumed a 30 min break between single escapes was more than enough to avoid fatigue accumulation. If fatigue affected performance, I assumed the effects were unbiased with regard to the comparisons of interest for this study. The order of the trials was randomized to eliminate confounding treatment (aquatic *vs.* terrestrial) with ecomorph, species, or individual experience.

## Kinematic analysis

Fast start performance data were collected using ImageJ software. Each video was analyzed independently by two technicians without prior knowledge of the species or ecomorphological hypothesis. Head, trunk, and tail measurements were taken once from each video when the animal was stationary, prior to the stimulus. Four timepoints were recorded from each trial. The initial position (position 0) was defined as the frame just prior to the first perceived movement. The subsequent three positions (positions 1–3) were the frames in which the trunk experienced maximum curvature during each of the first three lateral undulations (Fig. 2). The frame of maximum trunk curvature was determined by finding the frame with the minimum distance between shoulder and pelvis. Salamanders generate forward motion by generating a traveling wave down the trunk and tail when swimming or a standing wave in the trunk when walking (*Frolich & Biewener, 1992*). By recording the salamander's position at each maximum curvature of the trunk, we can determine the velocity generated by each wave. Here I focus on the net velocity generated by the first three waves, estimated as the linear distance between positions 0 and 3 (cm) divided by the time in seconds.

I used the mean net velocity across trials and observers as a single fast start performance metric for each individual in each medium. My goal was not to estimate maximal performance, but rather to compare performance between ecomorphs. Mean velocity across trials is highly correlated with the maximum of many trials, and unlike the sample maximum, the mean is unbiased and yields more reliable correlations with explanatory variables (*Adolph & Pickering, 2008*; *Head, Hardin & Adolph, 2011*).

For the purposes of this study, I assumed that velocity (distance from the attacker over time, regardless of direction) is a good indicator of ecological performance. Other components of escape behavior, such as changes in direction or speed, can be important for survival (*e.g.*, *Brodie III, 1992*). By focusing on the velocity of the initial burst, I intended to isolate basic physical ability from behavioral tactics. While direction was not quantified, most salamanders in this study tended to turn toward the center of the arena during aquatic trials, but moved directly forward during terrestrial trials.
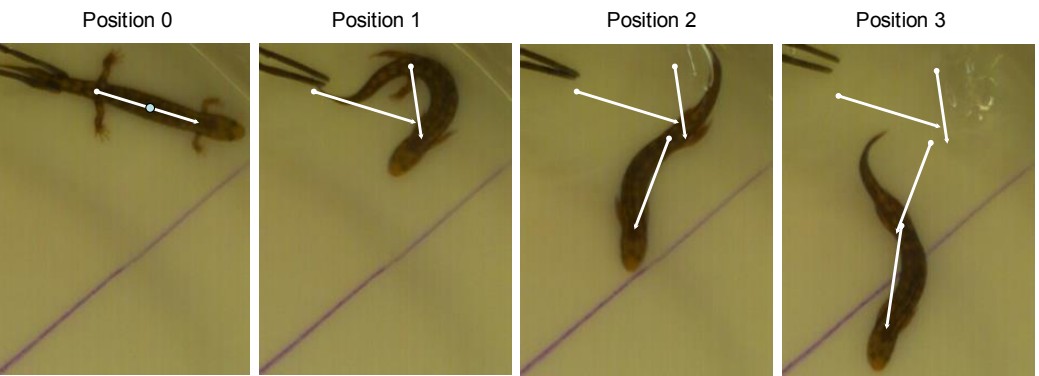

**Figure 2** **Measurement of fast start performance.** Video frames corresponding to each position in an aquatic escape trial were identified as the moment of maximum curvature of each locomotor wave. Total distance covered was measured between the midpoint of the line (axilla to groin) in positions 0 and 3. The specimen pictured is *Desmognathus marmoratus*, an aquatic ecomorph.

## Statistical analysis

To test for an overall signal of a trade-off, I estimated the Pearson correlation between terrestrial running velocity and aquatic swimming velocity expressed as body lengths per second. Further, to test whether the strength of the trade-off differs between northern and southern clades, I estimated separate correlation coefficients for each regional sample (Northern and Southern clades) and compared their 95% confidence intervals estimated *via* the normal approximation with the stats package in R (*R Core Team, 2024*).

To test whether the semi-aquatic and aquatic ecomorphs differed in performance, I used a mixed-effect linear model (*Fox & Weisberg, 2019*) with velocity as the dependent variable, species and trial habitat (aquatic or terrestrial) as factors, body length (measured from the snout to the groin), head length, and tail length as covariates, and individual as a random effect (because each individual was tested for both aquatic and terrestrial performance). I included all interaction terms, particularly a species by environment interaction term to test whether a performance difference between ecomorphs depended on the environmental context (aquatic *vs.* terrestrial). I then extracted the marginal mean velocity for each species x habitat combination and four planned contrasts to compare the species (different ecomorphs) within each clade under both aquatic and terrestrial conditions (*Lenth, 2025*).

Neither visual inspection of bivariate scatter plots nor the Shapiro–Wilk test of normality on the residuals indicated that any variable transformations were warranted ($W = 0.9856$, $p$-value $= 0.4418$). I used the variance inflation factor, VIF (*Fox & Weisberg, 2019*) to check for multicollinearity and found that head and tail length were highly correlated with body length. VIFs were acceptable and the model fit improved after eliminating head and tail length.

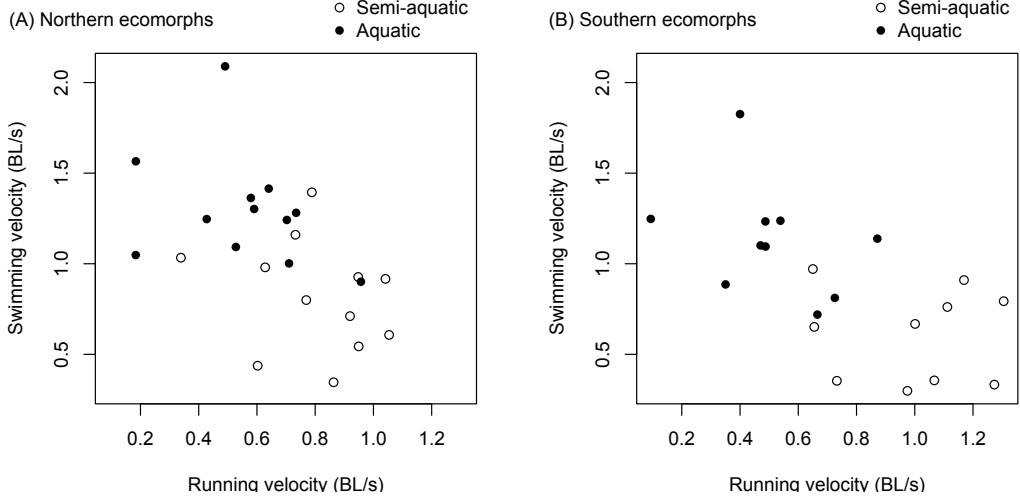

**Figure 3** **Trade-offs between terrestrial and aquatic escape performance.** Terrestrial (running velocity) and aquatic (swimming velocity) fast start speed expressed as body lengths per second were negatively correlated and different between ecomorphs (aquatic *vs.* semi-aquatic). The relationship was similar for ecomorphs in both the northern clade (A) and southern clade (B) of the *Desmognathus quadramaculatus-marmoratus* complex.

## RESULTS

There was a significant negative correlation between an individual salamander's fast start velocity in the terrestrial arena and its velocity in the aquatic arena ($r = -0.571$, $t_{42} = -4.507$, $p < 0.0001$). The relationship was similar for the northern and southern clades (Fig. 3), and separately estimated correlation coefficients were not significantly different (the 95% confidence interval was $-0.759$ to $-0.139$ for the northern clade, and $-0.834$ to $-0.245$ for the southern clade).

There was a significant interaction between environment and species for velocity during a fast start escape response (Table 1). The semi-aquatic ecomorphs were faster in the terrestrial arena, while the aquatic ecomorphs were faster in the aquatic arena (Fig. 3, Tables 2 and 3). Body length (snout to groin) and environment (aquatic *vs.* terrestrial) also had significant main effects (Table 1). Larger salamanders tended to be faster, and swimming velocity was faster on average than running velocity (Fig. 3, Table 2).

## DISCUSSION

The negative correlation between aquatic and terrestrial fast start velocity supports the hypothesis of a functional trade-off in *Desmognathus* salamanders. Moreover, performance differences between previously defined aquatic and semi-aquatic ecomorphs are consistent with a predictable evolutionary response to differences in the relative importance of aquatic and terrestrial locomotion that accompany shifts along the ecological gradient from large-stream habitat use to terrestrial foraging. This pattern is similar in evolutionarily distinct clades of *Desmognathus*, reinforcing the interpretation that parallel evolution of form, behavior, and function has accompanied adaptive diversification in the genus.

**Table 1** Mixed effect linear model for escape velocity (cm/s) of *Desmognathus* salamanders.

| Variable | $\chi^2$ | Df | *P*-value |
|---|---|---|---|
| Test habitat (aquatic *vs.* terrestrial) | 18.95 | 1 | <0.0001 |
| Species | 3.74 | 3 | 0.2904 |
| Body length | 28.24 | 1 | <0.0001 |
| Habitat:Species interaction | 72.19 | 3 | <0.0001 |
| Habitat:Body Length interaction | 0.40 | 1 | 0.5249 |
| Species:Body Length interaction | 0.58 | 3 | 0.9016 |
| Habitat:Species:Body Length | 7.14 | 3 | 0.0675 |

Notes.
Each variable was evaluated while accounting for all others (Type II tests) via the Wald $\chi^2$ Analysis of Deviance (*Fox & Weisberg, 2019*). Each salamander of each species was tested in both aquatic and terrestrial conditions (swimming *vs.* running). The marginal $r^2$ for the fixed effect portion of the model was 0.621.

**Table 2** Estimated marginal mean escape velocity (cm/s) of *Desmognathus* salamanders under terrestrial and aquatic conditions (running *vs* swimming).

| Clade | Species | Ecomorph | Habitat | Velocity | 95% CI |
|---|---|---|---|---|---|
| Southern | *D. amphileucus* | Semi-aquatic | Terrestrial | 5.76 | 4.67–6.84 |
| | | | Aquatic | 3.72 | 2.63–4.80 |
| | *D. aureatus* | Aquatic | Terrestrial | 2.90 | 1.89–3.91 |
| | | | Aquatic | 6.70 | 5.69–7.71 |
| Northern | *D. sp* | Semi-aquatic | Terrestrial | 5.10 | 3.88–6.33 |
| | | | Aquatic | 4.28 | 3.06–5.51 |
| | *D. marmoratus* | Aquatic | Terrestrial | 3.25 | 2.30–4.20 |
| | | | Aquatic | 7.58 | 6.63–8.52 |

**Table 3** Linear contrasts between marginal mean velocities of semi-aquatic and aquatic ecomorphs of *Desmognathus* salamanders within the southern clade (*D. amphileucus* and *D. aureatus*) and northern clade (*D. sp* and *D. marmoratus*).

| Habitat | Comparison | Difference | SE | $t_{df=72}$ | *p*-value |
|---|---|---|---|---|---|
| Terrestrial | *D. amphileucus–D. aureatus* | 2.86 | 0.745 | 3.841 | 0.0006 |
| Aquatic | *D. amphileucus–D. aureatus* | −2.98 | 0.745 | −4.003 | 0.0006 |
| Terrestrial | *D. sp–D. marmoratus* | 1.85 | 0.776 | 2.383 | 0.0198 |
| Aquatic | *D. sp–D. marmoratus* | −3.29 | 0.776 | −4.240 | 0.0004 |

Notes.
A positive difference means the semi-aquatic ecomorph was faster, and a negative difference means the aquatic ecomorph was faster. T-tests use the degrees of freedom determined by the Kenward-Roger method (*Lenth, 2025*). *P*-values were adjusted using the *Holm (1979)* sequential method to control family-wise error.

The running-swimming dilemma might be more evident in *Desmognathus* than in *Triturus* salamanders for two reasons. First, *Gvozdik & Van Damme (2006)* evaluated locomotor performance by chasing newts along a 200 cm runway and recording the maximum velocity per 25 cm interval. This is different from fast start velocity and measured over much longer interval than the data in this study (Fig. 2). Thus, we have measured different components of locomotor performance, and our different conclusions might reflect use of different variables rather than different animals. Second, *Triturus*

are chemically defended by potent neurotoxins and when threatened often adopt a static warning posture (the Unken reflex) rather than flight (*Brodie Jr, 1977*; *Telea, Stanescu & Cogalniceanu, 2021*). Thus, maximum speed might have little survival value for *Triturus*. On the other hand, *Desmognathus* are palatable and vulnerable to many predators (*Martof, 1962*; *Petranka, 1998*; *Sánchez-Hernández, 2020*; *Dempsey, Roden & Bidwell, 2021*), making escape ability an important contributor to fitness.

The significance of the functional parallelism between northern and southern clades depends to some extent on the resolution of ongoing efforts to understand the diversity and species boundaries in the northern clade. Parallel evolution in concert with ecological speciation is well understood (*Schluter & Nagel, 1995*; *Schluter, 2000*; *Nosil, 2012*). However, functional divergence between long-isolated species is not often matched by hybridizing species complexes or within-species polymorphisms because recombination disrupts coordinated inheritance of multiple loci affecting complex traits (*Dobzhansky, 1937*; *Felsenstein, 1981*; *Schluter & Rieseberg, 2022*). In this study, the functional differences within the northern clade, maintained in the face gene flow, appear to match those between long-isolated species in the southern clade.

## CONCLUSIONS

Trade-offs between aquatic and terrestrial locomotion are self-evident in broad comparisons (*Gvozdik & Van Damme, 2006*; *Kawano & Blob, 2013*; *Blob et al., 2016*), but not always relevant within species or ecologically conservative clades where other considerations dominate the observable variation (*Gvozdik & Van Damme, 2006*; *Nauwelaerts, Ramsay & Aerts, 2007*; *Zamora-Camacho, 2023*). *Desmognathus* salamanders are sometimes regarded as morphologically and ecologically conservative, but also as a textbook example of niche partitioning (*Hairston, 1994*). This study demonstrates that the ecological differences between the two most aquatic of the *Desmognathus* ecomorphs are accompanied by differences in fast start performance consistent with a functional trade-off forcing them into different evolutionary solutions to the running-swimming dilemma. This supports the general hypotheses that functional trade-offs contribute to ecological speciation and coexistence.

## ACKNOWLEDGEMENTS

Special thanks to D. R. Dittrich-Reed for contributing to the development and implementation of this project. K. Hamed assisted with field collections and C. D. Hulsey with high-speed video.

### Funding
The author received no funding for this work.

### Competing Interests
The author declares there are no competing interests.

## Author Contributions

- Benjamin M. Fitzpatrick conceived and designed the experiments, performed the experiments, analyzed the data, prepared figures and/or tables, authored or reviewed drafts of the article, and approved the final draft.

## Animal Ethics

The following information was supplied relating to ethical approvals (*i.e.*, approving body and any reference numbers):

University of Tennessee Institutional Animal Care and Use Committee protocol # 1870-0510.

## Field Study Permissions

The following information was supplied relating to field study approvals (*i.e.*, approving body and any reference numbers):

Virginia Department of Game and Inland Fisheries (# 37891) and the Georgia Department of Natural Resources.

## Data Availability

Code and data are available in the Supplemental Files.

## Supplemental Information

Supplemental information for this article can be found online at http://dx.doi.org/10.7717/peerj.20111#supplemental-information.

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
