# Peer review of "Trade-offs between aquatic and terrestrial locomotion and functional parallelism in Desmognathus salamanders"

_PeerJ, doi:10.7717/peerj.20111_

## Round 0.1 · original submission · Minor Revisions

Apologies for the long time that it took to review your manuscript. Both reviewers are enthusiastic concerning your study and have some (mostly minor) revisions. Please make sure that you redraft your manuscript making careful consideration of each suggested comment. I ask that you pay special attention to comments on the Methodology ensuring that the study could be replicated.

Reviewer 1 ·

Basic reporting

I commend the author for sharing the data and R script to increase transparency and reproducibility of the results. Not only is the R script provided, but it is also well annotated to increase readability of the different steps involved in the analysis. The manuscript is generally well-written, but would benefit from clarifying of the terminology used, further justifying the rationale for using only adults in the analysis, etc.
For instance,
1. “Ecological specialization” is a rather vague term that could be interpreted in a variety of ways, so it would be useful to define what is meant in this manuscript. As written, it appears that “variation in habitat preference” would be more accurate and specific for this study system. Futuyma and Moreno 1988’s definition of ecological specialization does not seem to be strongly consistent with the Desmognathus since their definition involves reproductive isolation whereas Desmognathus have extensive hybridization and introgression.
2. The historical definition of ‘ecomorphs’ by Ernest Williams involved convergent patterns between species that are not closely related. The definition has been relaxed since then, but it would be useful to report which definition is being used in the present study. Pyron et al. 2025 (https://academic.oup.com/sysbio/article-abstract/74/1/124/7848480) would be a useful reference since they discuss the black-bellied and shovel-nosed phenotypes and describe them as ecomorphs.
3. The use of ‘aquatic’ vs. ‘semi-aquatic’, ‘shovel-nosed’ vs. ‘black-bellied’, and ‘Marm’ vs ‘Quad’, respectively, needs to be consistently and systematically used in the manuscript, data, and R script. ‘Aquatic’ vs. ‘semi-aquatic’ would be the most appropriate dichotomy to use given the sampling method and the emphasis on trade-offs between aquatic vs. terrestrial environments. Moreover, it is incorrect to pool D. aureatus and D. marmoratus into the same ecomorph (‘marm’).

Experimental design

The author presents a compelling study system to delve into the factors that could affect gene flow between phylogeographic lineages and subsequently promote or constrain population differentiation across an environmental gradient and subsequently parallel evolution. Dusky salamanders (Desmognathus spp.) are a particularly valuable study system to investigate whether functional trade-offs in locomotor performance are present between syntopic species that partition themselves among different microhabitats within the same locality. The present study tests whether escape responses experience functional trade-offs (i.e., negative correlations) between aquatic and terrestrial conditions, using velocity as a metric of locomotor performance. While the premise for the study is compelling, the current state of the analyses does not sufficiently address the questions presented. Below, I describe recommendations to improve the current draft of the manuscript, but my primary concerns are as follows. First, the 12 individuals of semi-aquatic salamanders from VA need to be identified to species level. Second, there needs to be more careful use of terminology (e.g., ecomorph) and binning of the data (e.g., pooling species into a single ‘ecomorph’) to avoid potentially misleading results. Third, there needs to be stronger justification on the biological relevance of using adults rather than juveniles and the average velocity rather than the maximum instantaneous velocity, acceleration, or other metrics of locomotor performance.

For instance,
1. The “Species” column in the dataset is misleading because it is really a binary variable consisting of either Quad or Marm. First, change Quad and Marm to “semi-aquatic” or “aquatic”. Second, it would be better to run the analyses using the actual species names to determine whether there are enough similarities between the morphs of each clade to justify pooling them together into a single morph (‘aquatic’ for D. aureatus and D. marmoratus). Using separate analyses can increase the chance of Type I errors, and Bonferroni corrections were not applied to the results.
2. By focusing only the average velocity, the data analysis misses the opportunity examine other metrics of locomotor performance that could be easily calculated from the data already collected. For instance, maximum linear and rotational acceleration could be calculated since acceleration is often more important than velocity in determining the out of predator-prey interactions. In addition, locomotor performance could also be influenced by other parameters (e.g., maximum curvature coefficient, duration of the escape). Refer to Azizi and Landberg 2022 (https://doi.org/10.1242/jeb.205.6.841) for common metrics to assess escape performance.
3. The 12 semi-aquatic individuals from Little Laurel Creek need to be identified to the species level. GPS coordinates the localities samples in Pyron et la. 2022 (https://doi.org/10.1002/ece3.8574) are provided in the desmo900.csv file of the supplementary material, so the species of the 12 unidentified salamanders could be deduced based on their juxtaposition to the species from Pyron et al. 2022 or their location within a hybrid zone between two phylogeographic lineages or species.
4. What was the rationale for testing adults? Adults will often retreat to deeper burrows during the daytime and will only emerge at night when predators are unlikely to be present. In addition, the adults are most likely to have reached a size threshold where they are too large to fit in the mouths of predators (e.g., conspecifics), so it seems that juveniles would have been the most relevant life stage to examine trade-offs in locomotor performance. In addition, juveniles are typically the life stage that does the dispersing that contributes to habitat partitioning or gene flow.
5. It would be helpful to clarify why comparisons within clades used mixed-effects models to incorporate the three trials per environmental medium/condition, but then the net (summed) value of the velocities was used for the between-clade comparisons with a standard linear regression. If the LMM can account for repeated measures, then why not use one with Locality added as an additional fixed effect? If the choice was influenced by a relatively small sample size that does not have the statistical power to fit such a complex model, that should be stated.

Validity of the findings

The presence of functional trade-offs between aquatic and terrestrial conditions is well-justified from the correlation tests and can easily be replicated with the provided data and R script. The results from the ANCOVAs are also repeatable and match the patterns reported in the main text. It would be helpful to clarify the below information about other aspects of the statistical models, however:

1. Table 1: Mixed-effect Linear Models assume that there is no multicollinearity between variables, but head length and tail length are likely highly correlated with body length. Was Variance Inflation Factor or another metric used to test for multicollinearity in the independent variables? You could use car::vif().
2. Tables 1 - 3: the interaction between body length and ecomorph is also important to include since that the values reported in the methods section suggest that there are some size differences between the morphs and clades.
3. Tables 1 - 3: Was the aquatic group treated as the intercept for the statistical models? If so, please state to help with the interpretation of the results. Even though Tables 2 – 3 indicate that a positive coefficient means that a semi-aquatic ecomorph was faster and vice versa, it would be clearer to report the results from the table in a way that clarifies how you are making these conclusions (e.g., escape velocity in a terrestrial condition was 0.502 units lower in the semi-aquatic morph compared to the aquatic morph, Table 3). Also, report the coefficients of determination for your models to give an indication of how well the model describes your data.

Additional comments

Overall, I am enthusiastic about the study and feel that there is strong potential for this study to contribute valuable insights into ecological speciation, but further clarification and justification are needed as this time. Below, I describe specific comments on areas to improve.

Specific Comments
1. Lines 45 - 46: Divergent selection does not always need to specialization. Instead, the text should be corrected to reflect that contrasting suites of phenotypic traits confer higher fitness through divergent selection that could lead to performance trade-offs.
2. Line 48: The common names of fishes are also reported in sentence case.
3. Lines 112 – 113: Indicate how long the salamanders were acclimated/ quarantined in a laboratory setting before testing.
4. Lines 120 – 126: How long was food withheld from the salamanders prior to testing? Satiation can affect locomotor performance.
5. Line 139: Why were animals housed at 15 degrees Celsius but then tested at room temperature (~20 degrees Celsius)? These salamanders likely experience 20 degrees Celsius in their natural habitats, but it’s not clear why they were tested at a different temperature from their housing enclosure.
6. Line 141: How was the stimulus intensity regulated? There is substantial evidence from the literature indicating that escape responses can be affected by the strength of the stimulus, the direction that the stimulus was applied, and the ability of the animal to see the stimulus approaching.
7. Line 143: How was fatigue assessed? Did escape performance tend to decrease after each subsequent trial? A 30-min break may not be enough to recover from an escape response.
8. Lines 187 – 188: Why size-correct the velocity data for the northern vs. southern comparison, but not for the analyses conducted between medium/ environment within each clade? Including body length as a covariate is appropriate, so it’s not clear why the velocity data needed to be normalized to units of body length/ s.
9. Lines 197 – 198: Several consecutive nouns (‘noun trains’ can be confusing or difficult to interpret, so change “mean net fast start velocity” to “mean values for the net velocity during a fast-start escape response” for clarity.
10. Lines 219 – 221: Husak 2006 (https://www.jstor.org/stable/4139346) is a useful reference about how fast-start velocity is more relevant to survival than sprinting speed.
11. Figure 1: Specify that the aquatic morph is positioned closest to the scale bar in the photos and is represented by a specimen of Leurognathus marmoratus (now D. aureatus?), whereas the semi-aquatic morph is represented by a specimen of D. quadramaculatus (now D. amphileucus?)
12. R script, line 5: change QMescape.csv to peerj-113480-QMescape.csv.
13. R script, line 98: should rVND be changed to VNDr?
14. R script, line 111: should rVNW be changed to VNWr?

Reviewer 2 ·

Basic reporting

The manuscript is well written, in professional English, with only a few instances where additional clarification is needed (see general comments). The author has a clear hypothesis, describing the background and context well. Choice of experimental species is well-justified. Figures are clear, as are tables. Raw data is included.

Experimental design

The manuscript describes original research with the scope of PeerJ (fundamental biological sciences). The research question is well-defined, and uses appropriate methods and species to answer it. The study fills a gap in our understanding of performance tradeoffs by comparing replicated ecomorphs (aquatic and semi-aquatic salamanders) within and across distinct clades. The methods are described in sufficient detail to allow replication.

Validity of the findings

No comment (no concerns).

Additional comments

A few specific comments:

Line 139: you say that you allowed the salamander to acclimate to room temperature for at least 30 minutes – what was the room temperature? By how much did it differ from the holding temperature for the salamanders (said to be 15 C in line 120)? Did you measure the salamander’s temperature (e.g., by a non-contact IR thermometer) to verify that it had, in fact, equilibrated with room temperature?
Lines 151- 155: I find this to be confusingly described. What you call “positions” are, I think, more accurately described as “timepoints” (or “frames”). In addition, it would be clearer to specify that you are tracking the midpoints of the pelvis (midpoint of a line connecting the bases of the hindlimbs) and the shoulders (midpoint of a line connecting the bases of the forelimbs). Finally, line 151 says that “head, trunk, and tail measurements” were collected – this is in the same sentence where you talk about tracking positions, so it implies that head, trunk, and tail were being measured at each of the 4 timepoints. Is that what you meant to imply, or were the measurements taken once for each salamander, before the trial? If the latter, it would be better to break that out into its own sentence.
Lines 162 – 163: you say you used the mean net velocity across trials as the performance metric. It might be better to use the single fastest trial, if you are truly after maximal performance. One additional (potential) confound is the direction of the escape – if the salamander executes a sharp turn to one side, your velocity measurement is going to be lower than if the salamander had gone straight forward, even if the actual instantaneous velocity of the body was the same. An alternative might be to add up the distance traveled for each of the 3 “wavelengths,” and divide by the time taken to get the velocity. I can see arguments for both ways to calculate it, but I think the manuscript would be enriched by explicit discussion of why you concluded that your metric was the most appropriate.

---

## Round 0.2 · accepted · Accept

Thank you for the considerable time and effort you made in this draft! There two very minor comments that should be addressed before we can accept this manuscript for publication. Please see reviewer comments to make those adjustments.

Reviewer 1 ·

Basic reporting

Sufficient detail provided.

Experimental design

Expectations met.

Validity of the findings

Expectations met.

Additional comments

Thank you for the considerable time and effort you made in this draft! The revised manuscript is much improved and now contains sufficient detail and the appropriate analyses to address the aim of the study. The responses to the reviewer comments were well written and thoughtfully addressed the recommended edits. I have two minor edits to recommend, but I do not need to see a revised draft of the manuscript.

- Line 134: Include the citation for the statement made. Sass and Motta (2022) are often cited as justification for withholding food for two or more days to avoid the effects of satiation, but was based on fish that have much higher metabolic rates than salamanders. Typically, salamanders need at least seven days to reach a post-absorptive state. It is likely more accurate to state that waiting two days after a meal helped to prevent salamanders from regurgitating their food during the experiments, since they were probably still full and digesting at that point.
- Line 266: Do the two consecutive correlation coefficients for each clade correspond to the aquatic and semi-aquatic ecomorphs, respectively? It would be helpful to clarify which comparisons these coefficients correspond to.

Reviewer 2 ·

Basic reporting

The author did an excellent job responding to the criticisms and comments of the reviewers. I have no further concerns.

Experimental design

No concerns.

Validity of the findings

No concerns.

Additional comments

Great job!